# Assessment of Musculoskeletal Pain and Physical Demands Using a Wearable Smartwatch Heart Monitor among Precast Concrete Construction Workers: A Field Case Study

**Oscar Arias** *, **James Groehler, Mike Wolff** and **Sang D. Choi** *

Department of Occupational & Environmental Safety & Health, University of Wisconsin–Whitewater, Whitewater, WI 53190, USA

* Correspondence: ariasfo@uww.edu (O.A.); chois@uww.edu (S.D.C.); Tel.: +1-(262)-472-2194 (O.A.); +1-(262)-472-1641 (S.D.C.)

**Abstract:** This study aimed to quantify musculoskeletal symptoms/pain and characterize the physical demands at work and outside of work among precast concrete workers. Direct heart rate (HR) measurements and self-reported activity levels were used to estimate the physical demands. A total of 27 precast construction workers participated in a survey, and 21 wore a HR monitor smart watch for seven days. The HR data were parsed in minutes associated with occupational and nonoccupational physical activity. Correlation analysis and multivariate regression models were conducted to assess the associations between direct measured physical activity with self-reported physical activity, body mass index (BMI), years of work, smoking, and Borg ratings of perceived exertion (RPE). Approximately half of the participants experienced musculoskeletal symptoms in the last seven days and moderate functional limitations in carrying out activities of daily living (ADLs). The regression model revealed a positive relationship between direct measured moderate occupational physical activity (OPA) and the Borg RPE. Furthermore, an inverse association was found with BMI, smoking status, and years of work. The workers accrued a median of 415 min of moderate OPA per week. The findings showed a high amount of moderate OPA minutes per week and musculoskeletal issues among the precast concrete workers.

**Keywords:** musculoskeletal disorders; acute and chronic pain; survey questionnaire; workloads; construction industry; physical activity; heart rate; wearable electronic devices

## 1. Introduction

Construction work is a physically demanding job that exposes workers' musculoskeletal system to high mechanical loads. Daily work activities in construction include frequent lifting, carrying heavy loads, bending or twisting motions of the back, static work, exposure to vibrations, and extreme weather conditions [1]. These physical demands have been related to musculoskeletal disorders (MSDs) [2–6]. The prevalence of MSDs is high in the construction sector [7]. In 2015, there were 34.6 cases per 10,000 full-time equivalent workers. This rate was 16% higher compared to all industries combined [7]. Overexertion has been identified as the main cause of work-related musculoskeletal disorders (WMSDs) in the construction industry. Manual tasks involving lifting or lowering materials have been linked to approximately 30% of WMSDs and pulling, pushing, holding, catching, and carrying materials with an additional 37% of them [8]. Exposure to physical demands measured as moderate or vigorous occupational physical activity (OPA), including lifting heavy loads, carrying, pushing, and pulling, has been associated with an increased risk of work-related musculoskeletal disorders [9]. Conversely, moderate or vigorous intensity leisure time physical activity (LTPA) has shown positive health benefits to cardiovascular health [10–13].

Leisure time and occupational physical activity have contrasting health effects. While leisure time physical activity reduces cardiovascular mortality [14], OPA increases the cardiovascular disease mortality risk [15,16]. This contradictory effect is known as the "physical activity paradox". The differences in the frequency, intensity, duration, and recovery time seem to explain this [16,17]. These opposing effects require addressing OPA in physically demanding jobs to protect and promote workers' health. Construction workers seem to be exposed to a significant amount of OPA levels that might have a negative impact on their health [18,19]. For example, Hwang and Lee (2017) tested a wristband-type wearable heart rate sensor to continuously measure workers' physical demands during construction work and concluded that automated and continuous physical demand measurement can be enabled at the construction field [19].

Construction workers with musculoskeletal disorders have reported pain in different anatomical regions [9,20,21]. Musculoskeletal pain has been identified as an important burden for this working population that is aggravated by the physical characteristic of the job [5,6]. Work-related musculoskeletal disorders and pain have been identified as the main source of interference with normal work activities in the construction sector [7]. Furthermore, WMSDs have been identified as the leading cause of occupational disability in construction workers [22].

Hence, integrated interventions aiming at reducing the impact of the job's physical demands on construction workers' health are mandatory. Assessing physical activity levels at work as a proxy of physical demands is essential to designing tailored interventions addressing physical demands. Previous studies have collected information on physical activity based on self-reports, whereas other studies lack sufficient and accurate direct measurements [23,24]. Musculoskeletal pain is also an essential factor that needs to be accounted for during an intervention. Pain limits workers' ability to participate in work, social, and self-care activities. Significant pain also impairs quality of life [25]. The appropriate assessment of the physical demands and the presence of musculoskeletal pain in this working population is essential to help design effective interventions aiming at reducing the impact of the physical demands on the musculoskeletal system. A better understanding of the interaction between the physical demands of the work measured as OPA and the presence of musculoskeletal disorders and pain can shed light on addressing this goal successfully.

Precast concrete construction is an application of off-site construction (OSC) technique. Through this method, different nonstructural components are manufactured in a concrete facility and transported to the construction worksite for their installation [26]. Precast concrete workers typically engage in high physical demanding and intensive manual tasks and activities that can lead to work-related musculoskeletal injuries and disorders [27,28]. For instance, Abas et al. (2020) investigated the effect of work-related musculoskeletal disorders (WMSDs) among workers in a precast concrete fabrication facility. The precast concrete workers' musculoskeletal pain or discomfort were commonly in their upper back, shoulder, ankles/feet, and wrists/hands due to the manipulation of heavy loads, high force exertion, awkward working postures, static load, and repetitive tasks [29]. However, few or no studies examined occupational and nonoccupational physical activity associated with musculoskeletal symptoms/pain in the precast concrete construction industry.

This study aimed to (1) characterize self-reported and directly measured physical demands in a convenience sample of precast concrete construction workers, (2) characterize the presence of acute and chronic musculoskeletal pain in this target population, and (3) identify associations between our outcomes of interest and other variables. These results could help constitute a baseline for designing and implementing a tailored intervention aimed at reducing physical demands and promoting workers' health in the construction industry.

## 2. Materials and Methods

### 2.1. Subjects and Study Design

Construction workers from a precast concrete company located in the Midwest were invited to participate in this research study. This company manufactures and installs precast concrete components for the construction industry. For this study, we enrolled workers only from the manufacturing area. They were responsible for elaborating wall panels, beams, columns, and hollow core slabs. To be eligible for the study, participants had to be at least 18 years old and not taking any medication that could affect their heart rate. We consented and enrolled, from March to May 2022, a total of 27 construction workers. This group of workers performs similar tasks daily, so we consider that the information collected from them will provide us with a reasonable estimate of the prevalence of musculoskeletal symptoms, the presence of acute and chronic pain, and particularly allow us to estimate their physical demands, which are one of the most important factors associated with the occurrence of MSDs. The study materials and protocols were approved by the University of Wisconsin-Whitewater's institutional review board (IRB) (study number: FY2021-2022-88).

### 2.2. Self-Reported Musculoskeletal Pain, Pain Severity, Functional Limitation, and Perceived Exertion

The participants in the study were asked to complete a survey regarding job characteristics, the perceived exertion level at work, the presence of pain in six body areas (yes/no) that lasted less and more than three months in the past 12 months, the pain severity experienced in the last seven days, the treatment received to handle pain, and the pain-related functional limitations.

The self-reported perceived exertion for the last seven work days was quantified using the Borg ratings of perceived exertion (RPE) to provide an idea of the job's physical demands. We considered that self-reported physical demands during the last seven working days would provide us with a reliable estimate of their physical demands considering this group of workers performs very similar tasks daily.

Pain in six body areas was self-reported as a binary variable (yes/no) using a modified version of the Nordic questionnaire [30]. It was categorized as acute when it lasted between one day to three months and chronic if it lasted more than three months [31]. The pain severity during the last seven days was self-reported using a five-item questionnaire asking for the characteristics of the pain in the low back, neck/shoulder, wrist/forearm, knee, and ankle/feet. Each participant scored the severity (e.g., please rate the severity of the pain in your low back) using a 5-point Likert scale ranging from 1 (none) to 5 (extreme pain) [32].

The workers detailed the functional limitations by reporting their ability to perform ten daily activities over the past seven days (e.g., the ability to carry a shopping bag). Workers rated the levels of difficulty performing these tasks on a 5-point Likert-type scale, from 1 ("no difficulty in carrying out the task") to 5 ("unable to do a task without help") [33]. The scores allowed us to identify high difficulty levels in certain activities. We calculated the Cronbach alpha and Loveringer H to assess the internal consistency and scalability of the questionnaire assessing the pain severity and functional limitations.

### 2.3. Survey Questionnaire Apparatus Framework

The survey questionnaire was administered in person at the end of the seven days of heart rate data collection. A member of the research team distributed hard copies to the participants, who spent approximately 30 min completing them.

The final survey comprised the five domains below:

1. Demographic and general information;
2. Work characteristics, including perceived exertion (the Borg ratings of perceived exertion (RPE) scale);
3. Presence of acute and chronic musculoskeletal pain (modified version of the Nordic questionnaire). For example:

- "During the last three months, have you had pain or aching in any of the areas shown on the body diagram?", with the response options: lower back, shoulder, wrist or forearm, knee, neck, and ankles or feet;
- "In general, how much did this pain interfere with your normal work in the last seven days?", with the response options: not at all, a little bit, moderately, quite a bit, and extremely.

4. Pain severity in the five body regions over the last seven days was assessed using a five-point Likert scale (none, mild, moderate, severe, and extreme). For example:

- "Please rate the severity of the following symptoms that you may have experienced in the last 7 days. (a) Pain in your low back … "

5. Functional limitations to evaluate the ability to perform different activities were assessed using a five-point Likert scale. For example:

- "Do heavy household chores (some examples include washing walls or washing floors)?", with the response options: no difficulty, mild difficulty, moderate difficulty, severe difficulty, and unable to perform without help.

A group of experts reviewed the survey and performed content validation.

### 2.4. Work Description, Instruments, and Procedures

The primary precast concrete work was conducted inside a factory rather than a jobsite. Different crews have to manually set-up formwork consisting of lifting, measuring, cutting, placing side rails, and bulkheads. Included in this step is cleaning the bed through the use of scrappers and rags. Placing embedded material and pulling prestressed strands are also part of the formwork setup. Throughout the set-up, process workers have to bend, kneel, twist, and lift multiple items. Multiple subassembly groups weld and tie rebar reinforcing embedded material to be placed in the formwork before pouring the concrete. Once the form is set-up, the pouring takes place with the use of Tuckerbilt machines. Employees rake, vibrate, and screed the concrete. In certain elements, additional foam insulation is placed along with lifting devices and other embedded items. They then finish the concrete with the use of trowels, floats, and power trowel machines. Once the concrete is cured, it is then stripped from the bed through the use of indoor cranes. Part of this process is to use torches or bolt cutters to detention strand and also remove lifting cups with hand tools. Next, pieces are detailed and patched after stripping through the use of multiple hand tools, including torches, partner saws, scrappers, chisels, hammers, pry bars, and drills. Throughout the entire process, quality control checked the measurements, verified the concrete, and took samples.

The participants were asked to wear a smartwatch (Polar, Unite, Kempele, Finland) to continuously track their heart rate (HR) for seven consecutive days (working and nonworkdays). Other studies have used similar devices to quantify physical demands among construction workers [19]. This lightweight watch measures heart rate every second without causing significant discomfort to the workers. It uses a photoplethysmography (PPG) sensor to determine heart rate by quantifying the volume variations of the blood circulation in the skin. This sensor includes a light-emitting diode (LED) and a photodiode. The LED is a light source that illuminates the skin, and the photodiode measures the intensity of the reflected light from the skin. The reflected light intensity is inversely related to the blood volume variations. There is a low light intensity during heart muscle contraction corresponding to a high blood volume in the skin and high light intensity during heart muscle relaxation corresponding to a low blood volume in the skin. The volumetric variation of the blood flow allows for determining the heart rate [34]. Instructions on how to wear the watch were provided to each participant in the study.

The heart rate data were stored in a relational database management system (Microsoft SQL Server). The heart rate data were parsed into two sets: (1) seconds of heart rate levels associated with work alone and (2) seconds of heart rate levels outside of work. For each set of parsed data, each second was assigned to a physical intensity level according to

Norton's categories based on the %HRR [35]. The percentage of HR reserve (%HRR) is a relative measurement of physical demands considering individual differences [36,37]. Its calculation includes the heart rate while resting, or resting heart rate (RHR), and the heart rate at work (HRworking). Then, it establishes the percentage of the difference between working and resting heart rate and the heart rate reserve (difference between maximum heart rate and resting heart rate) [36]. Different intensity categories have been described based on the %HRR (sedentary: <20 %HRR, light: 20<40 %HRR, moderate: 40<60 %HRR, vigorous: 60<85 %HRR, and high: ≥85 %HRR) [35]. The %HRR is a calculation that measures the proportion of the maximum heart rate (HRMax) being used at a given point. For this calculation, we determined the resting heart rate (RHR), the heart rate while working (HRworking), and the maximum heart rate (HRMax). To obtain the resting heart rate, participants sat for 15 min and were encouraged to relax while the watch recorded the heart rate. During this time, the participants did not drink coffee or tea. The HRmax was calculated using Tanaka's formula: HRmax = 208 − 0.7 × age [38]. Each second of HR at work (HRworking) was categorized using the %HRR. The relative heart rate reserve %HRR was calculated using the following formula:

$$\%HRR = \frac{HRworking - RHR}{HRmax - RHR} \times 100\%$$

The total minutes accrued at each intensity category was calculated. This information allowed a determination of the number of minutes at each physical activity level per week. The data collected per individual were used to estimate the amount of physical activity at work and outside of work per week accrued at each physical activity level (e.g., sedentary, light, moderate, and vigorous). For calculating the physical activity at work, we assumed a week of five eight hour shifts. Physical activity outside of work was calculated assuming eight hours of physical activity per day seven days per week.

### 2.5. Statistical Analyses

Spearman's correlations were used to evaluate the associations between directly measured physical demands and self-reported physical demands, musculoskeletal pain, BMI, years of work, smoking status, and functional limitations. Spearman's correlations were chosen because of the highly skewed distributions of parameters and the scales used in the survey measures. The correlation statistical analyses explored the relationships between the directly measured moderate levels of OPA physical activity and parameters such as Borg RPE, self-reported minutes of moderate OPA, smoking, and years of work. In the multiple linear regression model, the dependent variable was directly measured moderate physical activity at work. The independent variables were BMI, self-reported Borg ratings of perceived exertion (RPE), self-reported minutes of moderate OPA, smoking, and years of work. All analyses were carried out in STATA 15.1 (StataCorp, College Station, TX, USA).

## 3. Results

### 3.1. Study Population

A total of 27 precast concrete workers participated in the study. They corresponded to 16.2% of the plant's total precast concrete working population. We collected survey data from 27 participants and heart rate (HR) data from 21 participants. Because of initialization errors of the watches, we lost the directly measured physical activity data on six participants. The sociodemographic characteristics of the participants are shown in Table 1. Most participants were Hispanic (45%) and White (45%) males. The participants' trades/occupations were cage tie assembler, hollowcore laborer, foreman, wetcast laborer, maintenance, patcher/detailer, quality control, and welder. They had a median age of 35 years old, a body mass index (BMI) of 29, which corresponds to being overweight, and 7 years of working experience in the construction industry, working approximately 10 h per day, more than five days per week. A total of 26% of the workers were current smokers. According to the Borg RPE scale, the self-reported exertion showed a median score of 14.

**Table 1.** Sociodemographic and work characteristics among the participants (*n* = 27).

| Sociodemographic and Work Characteristics | | |
|---|---|---|
| Sex, *n* (%) | | |
|     Male | 27 | 100 |
|     Female | 0 | 0 |
| Race, *n* (%) | | |
|     Black | 3 | 10 |
|     Hispanic | 12 | 45 |
|     White | 12 | 45 |
| Trade/Occupation, *n* (%) | | |
|     Cage tie assembler | 1 | 4 |
|     Hollowcore laborer | 1 | 4 |
|     Foreman | 3 | 10 |
|     Wetcast laborer | 16 | 60 |
|     Maintenance | 1 | 4 |
|     Patcher/detailer | 1 | 4 |
|     Quality control | 2 | 7 |
|     Welder/fabshop | 2 | 7 |
| Smoking people, *n* (%) | 7 | 26 |
| Age, *M* (range), years old | 35 | 19–54 |
| Body mass index (BMI), *M* (range), kg/m$^2$ | 29 | 19–35 |
| Years in construction work, *M* (range) | 7 | 0–28 |
| Hours of work/day, *M* (range) | 10 | 8–12 |
| Days of work/week, *M* (range) | 6 | 5–7 |
| Perceived exertion, *M* (range), 6–20 Borg RPE scale | | |
|     Current work shift | 14 | 11–17 |
| Self-reported occupational physical activity, *M* (range), min per week | | |
|     Moderate activity | 600 | 0–3600 |
|     Vigorous activity | 270 | 0–4200 |
| Self-reported physical activity outside of work, *M* (range), min per week | | |
|     Moderate activity | 210 | 0–4200 |
|     Vigorous activity | 10 | 0–4800 |

### 3.2. Self-Reported Physical Activity

The self-reported physical activity at work was higher than outside of work. At work, a total of 600 min/week corresponded to moderate physical activity and 270 min/week to vigorous physical activity. Outside of work, a total of 210 min/week corresponded to moderate physical activity and 10 min/week to vigorous physical activity (Table 1). The OPA levels reported were above the requirements of the general US guidelines for leisure time physical activity (150 min of moderate or 75 min of vigorous activity per week).

### 3.3. Musculoskeletal Pain

Approximately 57% of the participants reported acute musculoskeletal pain in the last 12 months. This proportion was higher than those reporting chronic musculoskeletal pain, which corresponded to 26%. The body areas more frequently associated with acute pain were the lower back (77%), knee (54%), and shoulder (54%). Likewise, the lower back (83%), knee (50%), neck (50%), and shoulder (54%) were the body areas mainly associated with chronic pain (Table 2).

A physician diagnosed 15% of the workers with MSDs during the previous 12 months. Additionally, 26% of the workers visited a health professional (e.g., chiropractor, physiotherapist, and physician) for treatment. Nearly 26% of the participants received a medication prescribed for their MSD symptoms. Furthermore, 15% of the workers had to implement changes in how they performed their work due to the fact of musculoskeletal pain, and 19% were required to take sick leave for the same reason (Table 2).

**Table 2.** Musculoskeletal pain and other musculoskeletal pain-related events among participants (*n* = 27).

| Musculoskeletal Pain in the Last 7 Days, Acute and Chronic Pain the Last 12 Months and Other Musculoskeletal Pain-Related Events | *n* | % |
|---|---|---|
| Musculoskeletal symptoms/pain | | |
| Reported musculoskeletal pain in the last 7 days * | 10 | 44 |
| Reported musculoskeletal pain in the last 12 months * | 13 | 57 |
| Body part/area: | | |
| Ankle or feet | 3 | 23 |
| Knee | 7 | 54 |
| Lower back | 10 | 77 |
| Neck | 5 | 38 |
| Shoulder | 7 | 54 |
| Wrist or forearm | 4 | 31 |
| Reported acute musculoskeletal pain in the last 12 months * | 13 | 57 |
| Body area: | | |
| Ankle or feet | 3 | 23 |
| Knee | 7 | 54 |
| Lower back | 10 | 77 |
| Neck | 5 | 39 |
| Shoulder | 7 | 54 |
| Wrist/forearm | 4 | 31 |
| Reported chronic musculoskeletal pain in the last 12 months * | 6 | 26 |
| Body area: | | |
| Ankle or feet | 2 | 33 |
| Knee | 3 | 50 |
| Lower back | 5 | 83 |
| Neck | 3 | 50 |
| Shoulder | 3 | 50 |
| Wrist/forearm | 1 | 17 |
| In the last twelve months | | |
| Visited a healthcare professional for musculoskeletal pain treatment | 7 | 26 |
| Reported a musculoskeletal disorder (MSD) diagnosed by a physician | 4 | 15 |
| Received medication due to the fact of musculoskeletal pain | 7 | 26 |
| Reported change in work habits due to the fact of musculoskeletal symptoms/pain | 4 | 15 |
| Reported sick leave due to the fact of musculoskeletal symptoms * | 5 | 19 |

* Differences in the subtotal population sample due to the fact of item nonresponse or missing.

More than 40% of the participants experienced musculoskeletal pain in the last seven days (Table 2). Among them, a substantial proportion reported moderate musculoskeletal pain in their low back (50%), and their arm, shoulder, or hand (40%). A lower proportion of workers (10%) reported severe pain in the low back and tingling in their arm, shoulder, and hand (Table 3). The reliability test for the musculoskeletal pain location and severity questionnaire reported a Cronbach alpha of 0.80, and the scale assessment reported a Loevinger H coefficient of 0.65.

**Table 3.** Pain location and severity among workers with musculoskeletal pain in the last 7 days (*n* = 10).

| Pain Location | Pain Severity | | | | | | | | | |
|---|---|---|---|---|---|---|---|---|---|---|
| | None | | Mild | | Moderate | | Severe | | Extreme | |
| | *n* | % | *n* | % | *n* | % | *n* | % | *n* | % |
| Pain in your low back * | - | - | 3 | 30 | 5 | 50 | 1 | 10 | - | - |
| Pain in your arm, shoulder, or hand * | 1 | 10 | 4 | 40 | 4 | 40 | - | - | - | - |
| Tingling ("pins and needles") in your arm, shoulder, or hand * | 5 | 50 | 2 | 20 | 1 | 10 | 1 | 10 | - | - |
| Pain in your legs or knees | 1 | 10 | 6 | 60 | 3 | 30 | - | - | - | - |
| Pain in your feet * | 3 | 30 | 3 | 30 | 3 | 30 | - | - | - | - |

* Differences in the subtotal population sample due to the fact of item nonresponse or missing. Cronbach α: 0.80; Loevinger H: 0.65.

### 3.4. Functional Limitations

As shown in Table 4, the participants who reported musculoskeletal pain in the last seven days experienced moderate limitations in performing activities of daily living, for example, recreational activities that involved some amount of force or impact on the upper limb (20%) and putting on shoes or socks (20%). They reported mild limitations principally for standing for one hour or more, reaching for an object on an overhead shelf, getting in or out of a car, stooping or bending over, kneeling or squatting, and when using a handheld tool or equipment (40%). The functional limitations questionnaire had a reliability test Cronbach alpha of 0.93, and the scale assessment reported a Loevinger H coefficient of 0.67 (Table 4).

**Table 4.** Functional limitations to perform activities of daily living (ADLs) among construction workers who reported musculoskeletal pain in the last 7 days (*n* = 10).

| Activity | Difficulty Level | | | | | | | | | |
|---|---|---|---|---|---|---|---|---|---|---|
| | None | | Mild | | Moderate | | Severe | | Unable to Do | |
| | *n* | % | *n* | % | *n* | % | *n* | % | *n* | % |
| Perform hefty household chores (for example, washing walls or washing floors) | 6 | 60 | 2 | 20 | 1 | 10 | - | - | - | - |
| Carry a shopping sack or basket/briefcase | 8 | 80 | 2 | 20 | - | - | - | - | - | - |
| Recreational activities that comprise some force or impact through your hands, arms, or shoulders (for example, golf, hammering, or tennis) | 5 | 50 | 3 | 30 | 2 | 20 | - | - | - | - |
| Stand for one hour or more | 6 | 60 | 4 | 40 | - | - | - | - | - | - |
| Reach for an object on an overhead shelf | 6 | 60 | 4 | 40 | - | - | - | - | - | - |
| Put on your shoes or socks | 5 | 50 | 3 | 30 | 2 | 20 | - | - | - | - |
| Get in or out of a car | 5 | 50 | 4 | 40 | 1 | 10 | - | - | - | - |
| Stoop or bend towards the floor | 5 | 50 | 4 | 40 | 1 | 10 | - | - | - | - |
| Kneel or squat | 5 | 50 | 4 | 40 | 1 | 10 | - | - | - | - |
| Use any handheld tool or equipment (some examples include a telephone, pen, keyboard, computer mouse, drill, hairdryer, or sander) | 6 | 60 | 4 | 40 | - | - | - | - | - | - |

Cronbach α: 0.93; Loevinger H: 0.67.

### 3.5. Directly Measured Physical Activity

The heart rate data were collected from 21 participants—177,266 s of OPA and 162,694 s of physical activity outside work were considered for the analysis. Each participant's directly measured heart rate data corresponding to OPA were adjusted to five working days of eight hours shifts per week. The heart rate data corresponding to physical activity outside of work was adjusted to seven days per week and eight hours per day. The median number of moderate and vigorous OPA minutes per week corresponded to 415 and 21 min, respectively. The median moderate and vigorous physical activity outside work was 257 and 10 min per week, respectively (Table 5).

**Table 5.** Median heart rate and minutes of directly measured occupational physical activity and physical activity outside of work during a week (*n* = 21).

| Physical Activity | Median | Range |
|---|---|---|
| Occupational physical activity (OPA) | | |
| Heart rate/min | 94 | 48–192 |
| Sedentary, min/week | 357 | 1–930 |
| Light, min/week | 1429 | 303–1839 |
| Moderate, min/week | 415 | 37–1280 |
| Vigorous, min/week | 21 | 0–897 |
| High, min/week | 0 | 0–7 |

**Table 5.** *Cont.*

| Physical Activity | Median | Range |
|---|---|---|
| Physical activity outside of work | | |
| Heart rate/min | 86 | 48–183 |
| Sedentary, min/week | 1268 | 365–2594 |
| Light, min/week | 1526 | 747–2333 |
| Moderate, min/week | 257 | 18–985 |
| Vigorous, min/week | 10 | 0–160 |
| High, min/week | 0 | 0–9 |

Sedentary (<20 %HRR), light (20<40 %HRR), moderate (40<60 %HRR), vigorous (60<85 %HRR), and high (≥85 %HRR).

The Spearman correlation between moderate levels of OPA and BMI, Borg (RPE), self-reported minutes of moderate OPA, smoking status, and years of work showed a negative correlation for BMI ($r_s = -0.4$), smoking ($r_s = -0.5$), and years of work ($r_s = -0.2$). A positive correlation was found with Borg ratings of perceived exertion ($r_s = 0.1$) and self-reported minutes of moderate OPA ($r_s = 0.2$). No statistically significant associations were found (Table 6).

**Table 6.** Spearman correlation coefficients and *p*-values between directly measured moderate levels of OPA and BMI, Borg (RPE), self-reported minutes of moderate levels of OPA, smoking, and years of work.

| Parameter | Directly Measured Moderate Levels (40<60 %HRR) of OPA | |
|---|---|---|
| | $r_s$ | *p*-Value |
| BMI | −0.4 | 0.2 |
| Borg (RPE) | 0.1 | 0.8 |
| Self-reported minutes of moderate OPA | 0.2 | 0.6 |
| Smoking | −0.5 | 0.09 |
| Years of work | −0.2 | 0.4 |

A multiple regression model was run to assess the relationship between directly measured minutes of moderate OPA per week and the variables BMI, Borg's ratings of perceived exertion (RPE), self-reported minutes of moderate OPA per week, smoking status, and years of work in the construction industry (Table 7). The model indicated a direct relationship between minutes of moderate OPA per week and the Borg RPE. For one unit increase in the score, there was expected an increase of 55 min in the total number of minutes of moderate OPA per week. Furthermore, an inverse association was found between minutes at moderate levels of OPA and BMI, smoking status, and the number of years of work. Per one unit increase in the BMI, there was expected a 21 min decrease in the total number of minutes at moderate levels of OPA. The expected reduction in the number of minutes for smoking and years of work was 395 and 13 min, respectively.

**Table 7.** Multiple linear regression model for directly measured minutes of moderate OPA per week.

| Outcome: Minutes of Moderate Occupational Physical Activity (40<60 %HRR) | | | | |
|---|---|---|---|---|
| Parameter | Estimate | 95% Confidence Limits | Standard Error | *p*-Value |
| BMI | −21.1 | −118.3–76.1 | 43.0 | 0.64 |
| Borg (RPE) | 55.1 | −162.7–273.0 | 96.3 | 0.58 |
| Self-reported minutes of moderate OPA | 0.0 | −0.2–0.2 | 0.1 | 0.7 |
| Smoking | −394.7 | −795.2–5.8 | 177.1 | 0.05 |
| Years of work | −13.4 | −67.9–41.2 | 24.1 | 0.6 |
| $F_{(5, 9)} = 1.56$, $p = 0.3$, $R^2 = 0.47$ | | | | |

## 4. Discussion

This study aimed to characterize the presence of work-related acute and chronic musculoskeletal pain and the physical demands measured as the physical activity levels at work among a group of precast concrete construction workers. According to our knowledge, this is the first study addressing these topics in this working population. Our long-term goal is to design and implement an integrated intervention to improve workers' health in this group of workers.

Our study's findings indicate that the precast concrete workers had a high prevalence of musculoskeletal pain (e.g., ~60% with acute pain and ~30% with chronic pain over the last 12 months). The most common body areas affected by acute pain were their lower back, knee, and shoulder. The areas most commonly affected by chronic pain were their lower back, knee, and shoulder. A number of participants (26%) received health assistance from medical professionals or took medications to manage their musculoskeletal pain in the last 12 months. Our study revealed that almost half of the participants had musculoskeletal pain during the previous seven days. Workers who experienced musculoskeletal pain in the last seven days reported mild functional limitations, such as standing for one hour or more, reaching for an object on an overhead shelf, getting in/out of a car, stooping/bending over, kneeling/squatting, and using a handheld tool/equipment. Precast concrete workers reported moderate limitations when performing recreational activities involving force or impact through their arm, shoulder, or hand and while putting on their shoes or socks. These findings are similar to the results from a previous study on a group of commercial construction workers who suffered acute and chronic pain at work that impaired them to perform their work and activities of daily living (ADLs) [20]. However, the prevalence of acute pain (86%) and the presence of pain in the last seven days (62%) were higher compared to this study. Previous studies reported a high prevalence of musculoskeletal disorders associated with pain and highly physically demanding tasks in the construction industry [39,40].

It is noted that, in the current study, the participants overreported minutes at work of moderate physical activity (600 min) compared to directly measured minutes of moderate physical activity (415 min). A previous study identified that the increased physical demands of a task could result in an increased overreporting of its duration. Furthermore, self-reported physical activity at work may incorporate the effect of psychosocial and other cognitive characteristics of the job [41]. Our direct measures of OPA indicated that the precast construction workers achieved a significant amount of moderate minutes per week (415 min). This result was higher than the findings reported among a group of commercial construction workers in Massachusetts (243 min) [18]. As expected, the data collected in this study indicated that this group of precast concrete workers was getting plenty of moderate OPA. The median amount of minutes accrued of moderate OPA was above the recommended leisure time physical activity guidelines of 150 min per week [42]. High OPA levels have been related to an elevated risk of cardiovascular disease (CVD) and all-cause mortality [43]. However, physical fitness seems to work as a modifier of this association. The workers with high physical fitness conditions and physical work demands did not experience an increased risk for CVD mortality [43,44]. The multiple linear regression model indicated that BMI, smoking, and years of work were negatively associated with directly measured moderate minutes of OPA. The smoking condition had the highest impact on the number of minutes of moderate OPA. Workers who smoke had a median reduction of 395 min in the number of minutes of moderate OPA. Smoking has been identified as a cause of lung cancer and cardiovascular and respiratory diseases [45]. Furthermore, epidemiological studies have established a negative impact of smoking on physical activity. According to them, there is an inverse association between smoking and physical activity [46–48]. An explanation for this finding is that positive and negative health behaviors tend to cluster together among individuals. Smokers are more likely than nonsmokers to adopt other risky behaviors that impair their health. Furthermore, education seems to work as a moderator of this association. Individuals with low education

tend to be most frequently physically inactive and smokers [49]. BMI and years of work had a lower impact on the median amount of OPA of 21 and 13 min, respectively. It is noted that the Borg RPE was associated with a higher number of moderate minutes of OPA. According to the model, per one unit of increase in the rating, there was a 55 min increase in the number of moderate minutes of OPA. The Borg's RPE appears to be a good estimator of the physical activity intensity level associated with construction work [50]. Although the Borg's RPE was subjective, and a high correlation existed between the 6 to 20 rating scale and heart rate during physical activity [50,51]. Workers in this study reported a median value of 14 (i.e., moderate-intensity activity level of "somewhat hard") for their Borg's perceived exertion (ranging from 11: intensity levels of "light" to 17: "very hard") [51]. The moderate intensity levels identified can be related to musculoskeletal pain and fatigue [52]. Additionally, high physical demands at work, with or without repetitive body movements, have shown to cause soft tissue damage that increases the risk of MSDs [53,54].

The main strength of our study was through the use of direct measures of heart rate data to calculate physical activity (our proxy for physical demands) over a 7 day follow-up period. Direct measures at work and outside work allowed us to identify variations in physical activity levels and to have a more reliable estimate of OPA. This information can be essential to manage physical demands within acceptable limits in this particular group of construction workers. A wearable electronic device (smartwatch heart monitor) has great potential for assessing the physical demands at the work site without interfering with the work tasks. Heart rate is a physiological variable that has been extensively used to quantify physical demands because of its association with cardiovascular loads [55]. The use of the %HRR allowed us to estimate the individuals' activity levels accounting for variations in heart rate levels due to the fact of individual characteristics, health conditions, and lifestyle factors. This relative measure focuses on relative changes in heart rate related to physical activities to infer the physical demands by accounting for individual heart rate differences [36,37]. Previous studies used %HRR on construction workers to characterize the physical demands among workers [19,56], concluding that this is a valuable tool to assess physical demands in the field.

Given the risk to workers' health, moderate or vigorous levels of physical demands should be restricted to a certain amount of time (e.g., risky if 40% HRR is maintained over 30–60 min, 60% HRR over 30 min, and 30% HRR over eight hours) [57]. Our results suggest the importance of implementing intervention programs to reduce the negative effect on workers' health of occupational physical demands and promote healthy lifestyles (e.g., limit occupational demands to <30%HRR, weight control, and tobacco cessation programs). A better understanding of the interaction between the job's physical demands and individual risk factors on workers' health could help develop tailored and integrated intervention programs. Reducing physical demands and promoting healthy lifestyles in the construction sector may significantly impact workers' health. Integrating approaches addressing work-related factors and healthy lifestyles have been proven successful and may serve as a guideline for intervention programs [58,59].

Some limitations require attention. The relatively small sample size (number of participants) limits the generalizability of our findings to only precast concrete workers with similar demographic characteristics and working practices. The heart rate data of six participants were not collected due to the presence of initialization issues during the HR data recording. However, given that this group of workers performs similar tasks each day, we did not foresee a significant impact on the estimation of the OPA during a working week. Another aspect that needs to be acknowledged is that the collected information allowed us to quantify only the workers' total physical demands at work. Designing interventions to reduce physical demands in the workplace requires identifying and assessing specific tasks associated with significant ergonomic risk factors as the focus of an intervention. We believe, however, that the current study's findings can serve as a baseline for future comparisons evaluating the impact of interventions on the total physical work demand. It is also worth mentioning that the effect measure shown by the model can

be overestimated, given the small sample size. However, it aimed to identify associations between variables related to moderate levels of OPA and not for inference purposes. One should note that our findings apply only to precast concrete workers with similar tasks and sociodemographic characteristics. Furthermore, recall bias can be of concern for questions referring to the events happening in the last 12 months, although we think musculoskeletal pain is a symptom most people can readily recall and report. Another limitation of the study was that we did not collect information on all risk factors that could be associated with musculoskeletal pain, such as psychosocial risk factors or diseases. Nevertheless, relevant factors related to MSDs were included in the study. To calculate the physical activity levels outside of work, we assumed that people perform 8 h per day during weekdays and weekends. This assumption might not be valid for some participants, particularly during weekends.

## 5. Conclusions

This study characterized the presence of acute and chronic musculoskeletal pain among precast concrete workers and directly measured their physical demands via wearable heart monitors to categorize OPA and physical activity outside work. The study demonstrated that the precast concrete workers who participated experienced acute and chronic musculoskeletal pain in their back, shoulders, and upper extremities while experiencing moderate functional limitations in performing activities of daily living (ADLs). A wearable heart monitoring technology, which has a great potential for assessing the physical demands at the work site and outside work, allowed to identify variations in the physical activity levels (e.g., %HRR categories) and to have a more reliable estimate of OPA. This information can be essential for managing physical demands within acceptable limits in this particular group of construction workers. Moreover, this study indicates that the construction workers who participated overreported self-reported amounts of moderate OPA compared to those of directly measured moderate OPA. A further investigation is warranted if the self-reported physical activity at work might capture many other cognitive and psychosocial aspects of the construction work. This study also reveals that directly measured moderate OPA had reasonable relationships with Borg's ratings of perceived exertion (RPE), BMI, smoking, and years of work. Finally, the findings from this study could help better understand how to improve integrated interventions and surveillance programs to lessen OPA and promote physical health outside work among construction workers.

**Author Contributions:** Conceptualization, O.A. and S.D.C.; methodology, O.A. and S.D.C.; resources, M.W.; data collection, J.G. and M.W.; data analysis, O.A. and J.G.; data interpretation, O.A., M.W. and S.D.C.; writing—original draft preparation, O.A. and S.D.C.; writing—review and editing, S.D.C., O.A. and M.W. All authors have read and agreed to the published version of the manuscript.

**Funding:** This research received no external funding.

**Informed Consent Statement:** Informed consent was obtained from all subjects involved in the study.

**Data Availability Statement:** Data sharing is not applicable for this study.

**Acknowledgments:** We would like to acknowledge José de Jesús Vargas Gutierrez for his valuable help with the Microsoft SQL server database management for the raw heart rate data analysis.

**Conflicts of Interest:** The authors declare no conflict of interest.

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
