# Peer review of "Assessment of Musculoskeletal Pain and Physical Demands Using a Wearable Smartwatch Heart Monitor among Precast Concrete Construction Workers: A Field Case Study"

_applsci, doi:10.3390/app13042347_

Round 1
Reviewer 1 Report
The research concerns a selected professional group: precast concrete construction workers and consists of the assessment of musculoskeletal pain and physical demands with the use of simple and effective tool: the wearable smartwatch heart monitor, which is very interesting idea. This is undoubtful important and interesting issue, but the small sample of participants and other limitations indicate that the research is rather of a pilot one, good introduction to the larger scale research program. Furthermore, the limitations described by the authors may have an important impact on the research results and should be taken into consideration in future research which plan presentation would be an added value.
The manuscript title corresponds to its content. The issues form a logical whole. The selected methods are adequate to the activities carried out. The research and utilitarian purpose are clearly defined. Obtaining the results of field research is to be used to reduce physical demands and promote healthy lifestyles, which is always important aim. The description of the results, their detail and discussion are satisfying. The manuscript has been written with great care. The selection of literature is correct, however, the legitimacy of citing publications older than 20 years should be considered.
Reviewer 2 Report
The manuscript entitled “Assessment of Musculoskeletal Pain and Physical Demands Using Wearable Smartwatch Heart Monitor Among Precast Concrete Construction Workers: A Field Case Study” is an important work of promoting the health of workers in the construction industry. However, the manuscript needs to be improved, and much more information about the procedure must be added to allow replication of the study.
1. How is pain level measured in the design of the questionnaire. Is this concept a subjective pain perception or an objective degree scale. Are there any relevant descriptions and guidelines in the questionnaire. Whether the subjects can correctly determine their own pain level by the content of the questionnaire. And if accuracy cannot be guaranteed, whether the results of the questionnaire are limited. (The description of the functional limitations section is more detailed in comparison.)
2. The authors must describe in detail the form of questionnaire distribution (online or offline, and if online, what is the distribution medium), the specific time of questionnaire distribution (the start and end of questionnaire collection), and other content.
3. The design of Tables 1 and 2 (headers, data, units) is not very clear.
4. The need for wearable heart monitoring technology is mentioned in the conclusion, so I think more should be said about smartwatches in the study methodology.
Reviewer 3 Report
This study aimed to quantify musculoskeletal symptoms/pain and characterize the physical demands at work and outside of work among precast concrete workers. I have few points to mention for improving this research:
1. Sample size is very low, also it is not justified for this type of study?
2. The data collection is not appropriate for characterizing the physical demands. It must be done activity wise. It is not clear from the current sample description.
3. The study span is too small for this type of data collection.
Round 2
Reviewer 2 Report
congratulations!
Reviewer 3 Report
The authors have identified the issues raised by the reviewers. However, in my opinion there is room for improving the manuscript. Also, if certain things are not present in the research, then how it will fill the certain research gaps.
Round 3
Reviewer 3 Report
I have no further comments.